# Understanding and Tackling Scattering and Reflective Flare for Mobile Camera Systems

Fengbo Lan
fengbo.lan@connect.polyu.hk
The Hong Kong Polytechnic University
Hong Kong, China

Chang Wen Chen
changwen.chen@polyu.edu.hk
The Hong Kong Polytechnic University
Hong Kong, China

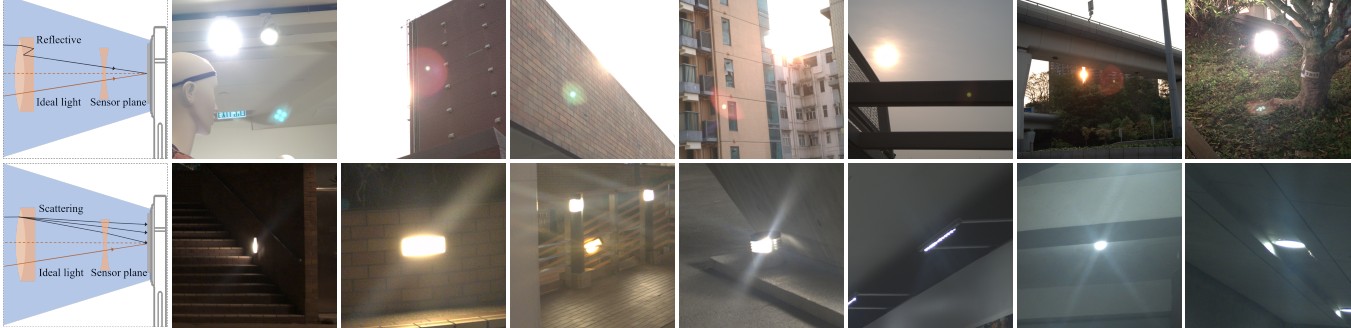

Figure 1: Reflective and Scattering Flares in the Dataset. The dataset includes examples of both reflective flares (top), which occur due to internal reflections within lens systems, and scattering flares (bottom), caused by light scattering from microscopic imperfections like dust or scratches on lens elements.

## Abstract

The rise of mobile devices has spurred advancements in camera technology and image quality. However, mobile photography still faces issues like scattering and reflective flares. While previous research has acknowledged the negative impact of the mobile devices' internal image signal processing pipeline (ISP) on image quality, the specific ISP operations that hinder flare removal have not been fully identified. In addition, current solutions only partially address ISP-related deterioration due to a lack of comprehensive raw image datasets for flare study. To bridge these research gaps, we introduce a new raw image dataset tailored for mobile camera systems, focusing on eliminating flare. This dataset encompasses over 2,000 high-quality, full-resolution raw image pairs for scattering flare, and 1,200 for reflective flare, captured across various real-world scenarios, mobile devices, and camera settings. It is designed to enhance the generalizability of flare removal algorithms across a wide spectrum of conditions. Through detailed experiments, we have identified that ISP operations, such as denoising, compression, and sharpening, may either improve or obstruct flare removal, offering critical insights into optimizing ISP configurations for better flare mitigation. Our dataset is poised to advance the understanding of flare-related challenges, enabling more precise incorporation of

flare removal steps into the ISP. Ultimately, this work paves the way for significant improvements in mobile image quality, benefiting both enthusiasts and professional mobile photographers alike.

## CCS Concepts

• **Computing methodologies**; • **Artificial intelligence**; • **Computer vision**; • **Image and video acquisition**; • **Computational photography**;

## Keywords

image processing pipeline, flare removal, raw image dataset

## 1 Introduction

Lens flare [12, 15, 17, 22] is a common optical artifact degrading image quality and visual appeal. It occurs when stray light enters the camera lens and interacts with the imaging sensor. This phenomenon is particularly prevalent in mobile computational imaging due to several factors. Firstly, mobile cameras often utilize plastic lenses, which generally exhibit lower quality compared to professional-grade glass lenses. Secondly, the cost constraints of mobile devices often preclude the inclusion of expensive anti-reflective (AR) coatings [5], further exacerbating the issue of lens flare.

Early approaches to flare removal relied on image processing techniques [3, 6, 16, 25]. However, recent advancements in deep learning and the availability of flare datasets [8–10, 29] have spurred the development of learning-based methods for tackling this problem. Notably, researchers have identified tone-mapping as a critical factor affecting restoration performance [8, 32]. Tone-mapping is a non-linear and non-invertible process within the image signal processing (ISP) pipeline that can hinder accurate flare removal.

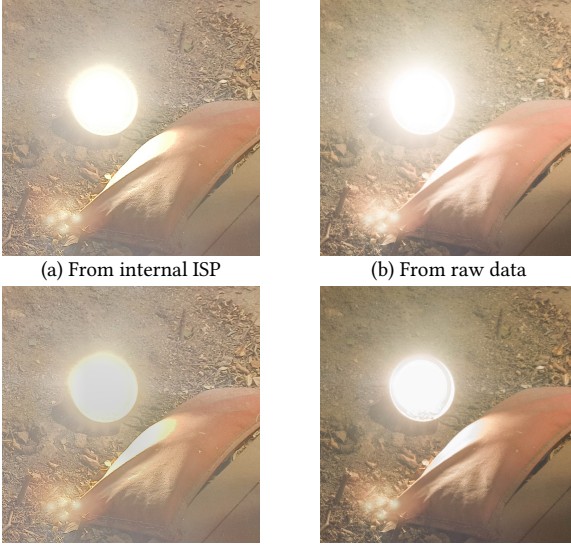

(a) From internal ISP                    (b) From raw data

(c) Highlight recovery from (a)          (d) Highlight recovery from raw data

**Figure 2: Images processed by a mobile camera built-in image signal processing (ISP) pipeline, like (a), lose significant detail compared to raw images (b). This information loss becomes evident when attempting to recover details in highlights, as shown in (c) and (d). The ISP-processed image (c) reveals far less detail than the image processed from raw data (d).**

However, the ISP pipeline encompasses multiple non-invertible transformations beyond tone-mapping, each contributing to the loss of original image information, as exemplified in Fig. 2. For instance, denoising algorithms, while essential for mitigating noise amplified by high ISO settings and small sensor sizes (especially in nighttime photography), inevitably remove some image details as a trade-off. Additionally, user preferences often drive manufacturers to incorporate sharpening operations that enhance image details, potentially introducing artifacts. Finally, image compression, typically into lossy formats like JPEG, discards information to achieve manageable file sizes. These observations raise crucial questions regarding the impact of various ISP steps on flare removal:

*Do other non-invertible ISP operations, such as denoising, sharpening, and compression, hinder or benefit the effectiveness of flare removal? Does their influence vary for different types of flares (e.g., scattering vs. reflective)?*

Understanding the interplay between these operations is vital for optimizing the ISP pipeline and effectively tackling lens flare. Unfortunately, the lack of readily available raw image datasets has limited previous research to mimicking individual operations rather than conducting comprehensive investigations.

This paper addresses these challenges by introducing a novel dataset and methodology specifically designed for lens flare removal. Our approach leverages the unique properties of raw images to decompose the non-linear, non-invertible processing steps within the ISP for detailed analysis. The dataset comprises over $2,000$ high-quality, full-resolution raw image pairs exhibiting scattering flare and $1,200$ pairs with reflective flare, all captured using mobile

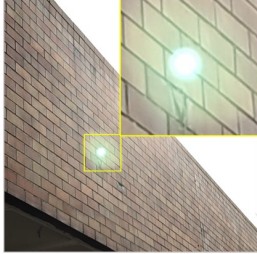 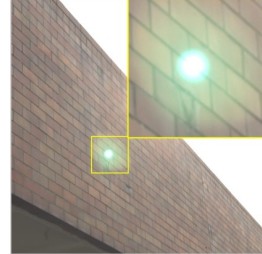

(a) From internal ISP                    (b) From raw data

**Figure 3: (a) Image processed by a mobile camera built-in ISP. (b) Image processed from raw data using an external pipeline. Mobile camera ISPs often apply sharpening and compression, which can introduce artifacts, as seen in (a).**

phone cameras. Each pair includes both the raw image and its processed counterpart from the internal ISP, providing rich and diverse data for training and evaluation. To the best of our knowledge, this is the first dataset offering raw image data for both scattering and reflective flare removal. Our experiments reveal that existing approaches, primarily trained on images with localized flare artifacts, may not generalize well to large-scale, global flare effects. By including both local and global flare corruptions, our dataset enhances the generalizability of these methods. This paper makes the following key contributions:

- Introduce a unique raw image dataset specifically curated for lens flare removal, featuring over 3,000 real-world examples that address the limitations of existing datasets by including diverse lighting and environmental conditions.
- Enhances the generalizability of lens flare removal techniques by incorporating a comprehensive range of both local and global flare effects.
- The impact of non-invertible ISP operations (denoising, compression, sharpening) on flare removal is thoroughly investigated, revealing their varying effects.
- A comprehensive analysis of the interplay between flare removal and the ISP pipeline is provided, offering valuable insights for effective integration.

## 2 Related Works

This section presents an overview of the flare removal problem and current available datasets, followed by a discussion on the utility of raw image datasets in mobile computational photography.

### 2.1 Flare Removal Problem

Lens flare, a common issue in photography, arises from various sources such as light scattering within the lens system, reflections between lens elements, and the impact of dust, contaminants, or scratches on lens surfaces. It can significantly affect the quality of photographs by introducing unwanted artifacts. Lens flare is broadly categorized into two types: scattering flares and reflective flares, each with distinct characteristics and appearances.

Scattering flares occur due to light interacting with microscopic imperfections within the lens system, leading to light scattering in different directions and resulting in visible artifacts such as veiling

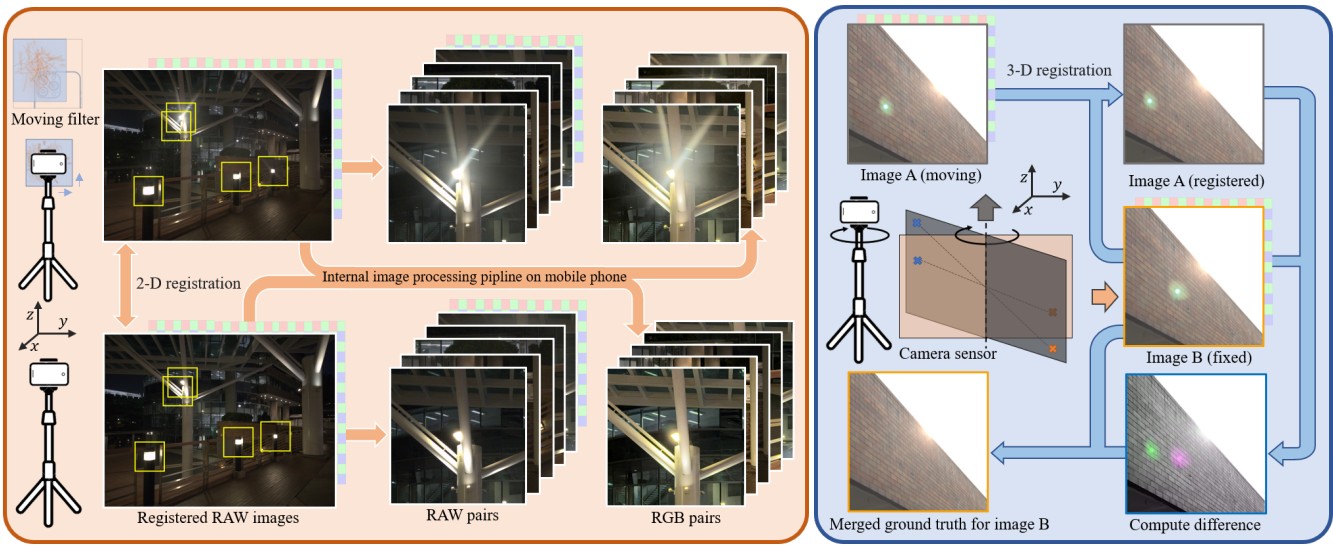

(a) Capturing scheme for scattering flare images  (b) Capturing scheme for reflective flare images

**Figure 4: Pipeline for Capturing Scattering and Reflective Flare Image Pairs. (a) Scattering Flares: A camera filter with varying degrees of stain simulates real-world lens imperfections. Paired images are captured, aligned using 2D registration, and cropped to isolate the flare region. This results in paired patches containing flare-corrupted and clean image data. (b) Reflective Flares: Exploiting the symmetrical nature of reflective flares, we capture two images with slightly different camera orientations. 3D registration is then used to calculate the difference between the images, which represents the flare itself. These images are then merged to create a ground truth image without flare.**

glare or a series of distortions in the image [13, 19, 21, 24]. The appearance of these flares depends on the distribution and nature of the imperfections. For example, dust on the lens surface may create bright spots or streaks, while scratches can lead to elongated artifacts. Multiple defects can result in complex overlapping flare patterns, degrading the image further.

Reflective flares, on the other hand, are the result of light reflecting between lens elements in multi-element lens systems [6, 16, 18, 25]. These internal reflections can produce geometric shapes such as concentric rings or polygons, depending on the lens design and the positioning of the light source. Reflective flares are especially noticeable when the light source is near the optical axis or the lens comprises many elements. Furthermore, these flares can vary in shape and clarity based on their focus. In-focus reflective flares appear as sharp, defined patterns, whereas out-of-focus flares often manifest as diffuse, irregular shapes, influenced by lens design and aperture shape.

## 2.2 Overview of Flare Datasets

Several significant datasets have been developed for flare removal research. Wu *et al.* introduced a dataset featuring both captured flare-only images and simulated scattering flare images [29]. Qiao *et al.* [20] collected unpaired flare-corrupted and flare-free images, suitable for networks that capture distribution differences but unsuitable for pixel-to-pixel neural networks due to the lack of paired data . The Flare7K dataset [8] includes synthesized scattering and reflective flare images, alongside real images for evaluation. Dai *et al.* later expanded this dataset to include real scattering flare images captured in a darkroom, addressing some limitations of their

earlier datasets [9]. Zhou *et al.* proposed a dataset collected using electronic devices for evaluation purposes [32].

Despite these contributions, the issue of reflective flare, which is common in everyday photography, remains underexplored. Creating ground truth data for reflective flare is challenging due to its inherent presence in optical systems. Flare7K and Wu *et al.*'s datasets simulate reflective flares at specific angles without considering the image content context, such as the shape of the light source [8, 29]. The Bracket Flare dataset by Dai *et al.* addresses in-focus reflective flare with a night-time dataset using a novel composition method [10].

Our dataset offers a comprehensive resource for studying both reflective and scattering flares, covering in-focus and out-of-focus reflective flares, as well as local and global corruption in scattering flares. This diverse collection aims to deepen and broaden the scope of flare removal research.

## 2.3 Image Signal Processing and RAW Data

The Image Signal Processing (ISP) Pipeline is a critical component in digital cameras, specifically engineered to process the complex data captured by camera sensors. The main objective of an ISP is to transform raw sensor data into a visually appealing image format, such as JPEG [26]. This transformation process encompasses several steps, starting with a linear transform that includes demosaicing [14], white balancing [2], and color correction [30]. Subsequent steps involve non-linear transformations, such as denoising [7], tone mapping [11], and JPEG compression [26]. Additionally, in the realm of mobile photography, ISPs are tasked with advanced image enhancement techniques, like sharpening, to compensate for the

**Table 1: Detailed specifications of the mobile phones used for constructing the dataset.**

| Model | Manufacturer | CMOS sensor (Main camera) | Specification | Released year |
|-------|-------------|---------------------------|---------------|---------------|
| iPhone 13 | Apple | IMX603 | 12 MP sensor, 1/1.7-inch sensor, 1.7$\mu$m pixels, 26 mm equivalent f/1.6-aperture lens | 2021 |
| Pixel 7 | Google | GN1 | 50MP sensor, 1/1.31-inch sensor, 1.2$\mu$m pixels, 24mm equivalent f/1.85-aperture lens | 2022 |
| iQoo Neo 7 | Vivo | IMX766V | 50MP sensor, 1/1.56-inch sensor, 1$\mu$m pixels, 23mm equivalent f/1.88-aperture lens | 2022 |
| Find X6 Pro | OPPO | IMX989 | 50MP sensor, 1-inch sensor, 1.5$\mu$m pixels, 23mm equivalent f/1.8-aperture lens | 2023 |

**Table 2: Statistics of the collected data.**

| Data | Scattering | Reflective |
|------|-----------|-----------|
| Indoor | 701 | 79 |
| Outdoor daytime | 0 | 803 |
| Outdoor nighttime | 1326 | 366 |
| Total number | 2027 | 1248 |

limitations imposed by smaller sensors and the inherent processing pipeline, as illustrated in Fig. 3.

## 2.4 Comparison with Existing Datasets

While existing datasets serve as essential resources for addressing the flare removal challenge, they fall short in providing the raw image data necessary to fully comprehend the impact of various ISP operations. This gap is particularly noticeable in operations like tone mapping, which significantly influence performance. Zhou *et al.* highlighted this issue, pointing out that methods like those employed in the Flare7K dataset [8], which combine flare and scene images in a gamma-corrected space, fail to account for the non-linear nature of tone mapping, leading to synthetic images with unrealistic contrast and color distortions, especially around bright light sources [32]. In the absence of paired raw data for flare-affected and clean images, these researchers have resorted to simulating tone-mapping effects within their synthesis pipeline through a pixel illuminance-based weighting scheme. While this approach represents a creative attempt to mimic the non-linear effects of tone mapping, it underscores the critical need for access to genuine raw data. This need is not only for improving the accuracy and generalizability of flare removal models but also because raw images are subject to a wider range of operations beyond tone mapping. Our proposed raw image dataset, therefore, presents a unique and valuable resource for examining the influence of tone mapping and other ISP operations on the manifestation of lens flare artifacts, setting the stage for more advanced research in the field.

## 3 Dataset Construction

This section details the creation of our dataset, covering the capturing devices and settings, followed by specific capturing schemes for scattering and reflective flare.

### 3.1 Capturing Settings

We employed mobile phone models from various manufacturers as capturing devices to avoid potential similarities in lens flare within

the same series. The chosen devices, detailed in Table 1, are popular in mobile photography and represent a range of capabilities. For consistency, we used the main camera of each device with manual control over exposure and focus, whenever possible. Multi-frame fusion was disabled to ensure the best raw image quality. The statistics of the collected data is tabulated in Table 2.

### 3.2 Scattering Flare

To simulate a lens with defects, we introduced a stain-corrupted camera filter in front of the capturing devices. Varying the filter's location relative to the camera simulated different levels of corruption due to the varying defect levels across the filter. While the camera remained fixed on a tripod, minor vibrations during capture could cause misalignment between the flare-corrupted and flare-free image pairs. To address this, we performed sub-pixel registration using SURF feature extraction and matching [4]. Given the small movements, only translations along the vertical and horizontal axes were computed. The registration process was first applied to internally processed images and then converted for the raw image data with its integer pixel grid. For each registered pair, we identified areas with light sources and applied a detection algorithm to pinpoint their positions, as illustrated in Fig. 4a. Both raw and internally processed images were cropped into patches centered on these light source positions. The raw patches were then processed externally to generate high-quality RGB image pairs. Finally, before inclusion in the dataset, predefined metrics were used to filter out low-quality pairs, ensuring the overall dataset quality.

### 3.3 Reflective Flare

To capture reflective data by utilizing the symmetrical property between the flare and the light source, we employed the following method. As illustrated in Fig. 4b, we rotated the camera to change the light source position, which resulted in a shifted flare image. Through image registration and warping, we aligned the two captured images. We then used image subtraction to identify the flare location and merged the images. This allowed us to use the corresponding area from one image to compensate for missing information due to the flare in the other. Repeating this process with the roles reversed for the two images provided two pairs of flare-corrupted and flare-free images from a single capture.

Camera rotation and image registration necessitate image warping and interpolation. Direct interpolation on raw image data can introduce artifacts like color aliasing and disrupt the noise model.

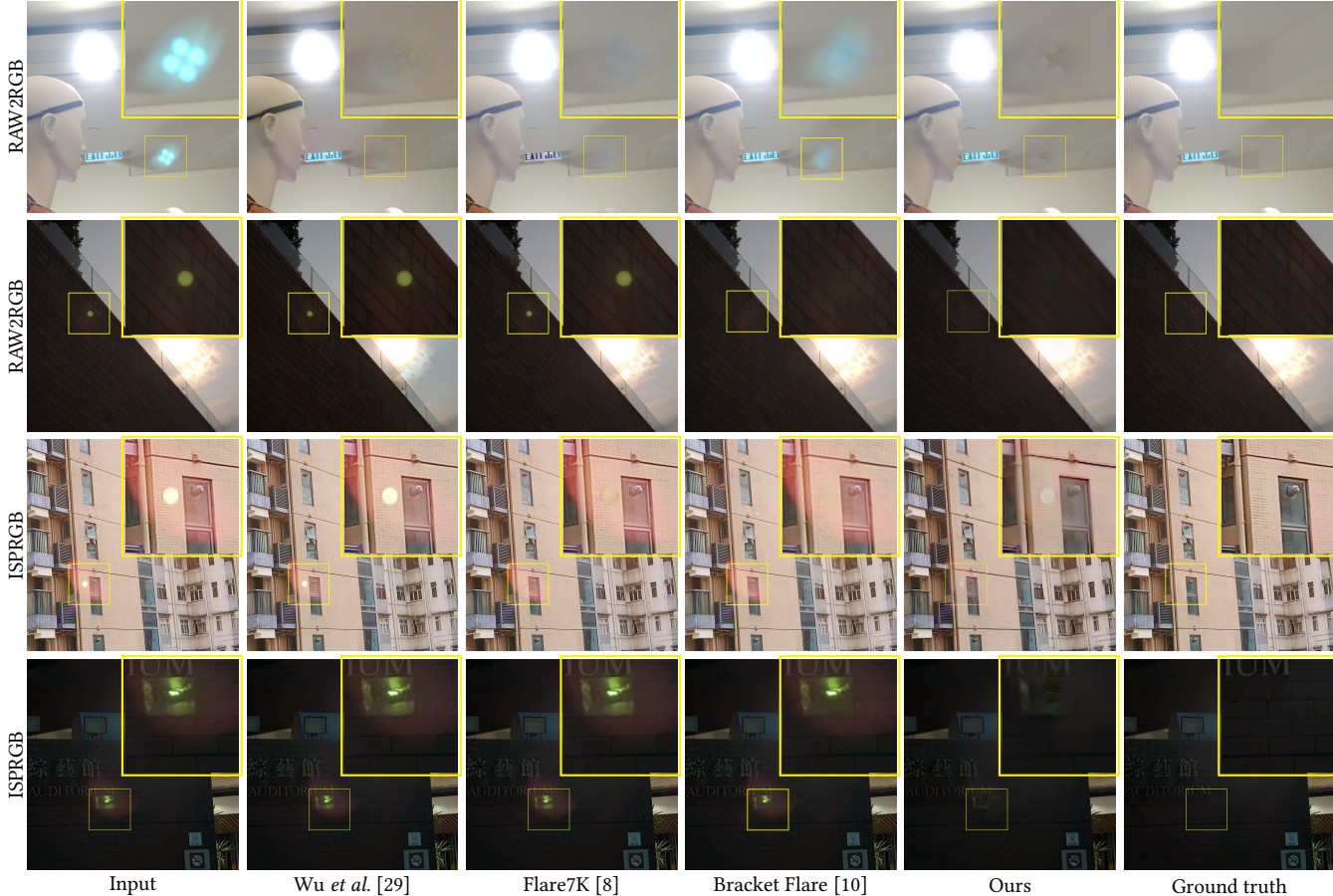

**Figure 5: Visual comparison of reflective flare removal using different schemes.**

To avoid this, we provide the original raw images and perform registration only on the internally processed and externally generated RGB images.

## 4  Experiments

This section evaluates the performance of flare removal models using images processed by both the internal ISP of a mobile phone and an external processing pipeline on a computer. We explore the differences in performance between these two processing methods by analyzing the impact of each processing step.

### 4.1  Data Preparation

Our evaluation includes both raw images and those processed by the mobile phone's internal ISP. For raw images, we convert flare-corrupted images and their corresponding ground truth pairs into RGB images using a custom MATLAB pipeline, *RAW2RGB*. This pipeline performs black level correction, white balancing, demosaicing, and color space conversion, but excludes denoising or post-processing techniques like sharpening or compression. These images are referred to as *RAW2RGB*. Images processed by the internal ISP are denoted as *ISPRGB*. Both scattering and reflective flare images undergo these processing steps.

For scattering flares, we utilize a light source detection algorithm to identify light sources and crop the high-resolution raw image pairs into $512 \times 512$ flare-corrupted pairs. This detection algorithm, adapted from [29], includes enhancements to reduce background misclassification in images with prominent white areas. We manually mask the background in each ground truth image to improve accuracy, use morphological operations to refine the detection masks, and employ a multi-light source detection approach to better represent complex real-world scenes. For reflective flares, we prepare pairs at a resolution of $1024 \times 1024$. Both *RAW2RGB* and *ISPRGB* datasets are processed in this manner.

### 4.2  Evaluations on the Dataset

*4.2.1  Experiment Settings* We assess the performance of cutting-edge flare removal techniques. Training for scattering flares follows the network settings from prior studies [8, 10, 29]. Since the model from Wu *et al.*is unavailable, we train a new model using their released code and data. For the Flare7K [8] and Flare7K++ [9] datasets, we evaluate the pre-trained Uformer model [28]. Training for reflective flares involves models from Flare7K [8] and Bracket Flare [10], using our collected data for separate treatments of scattering and reflective flares. Results are documented in Table 3.

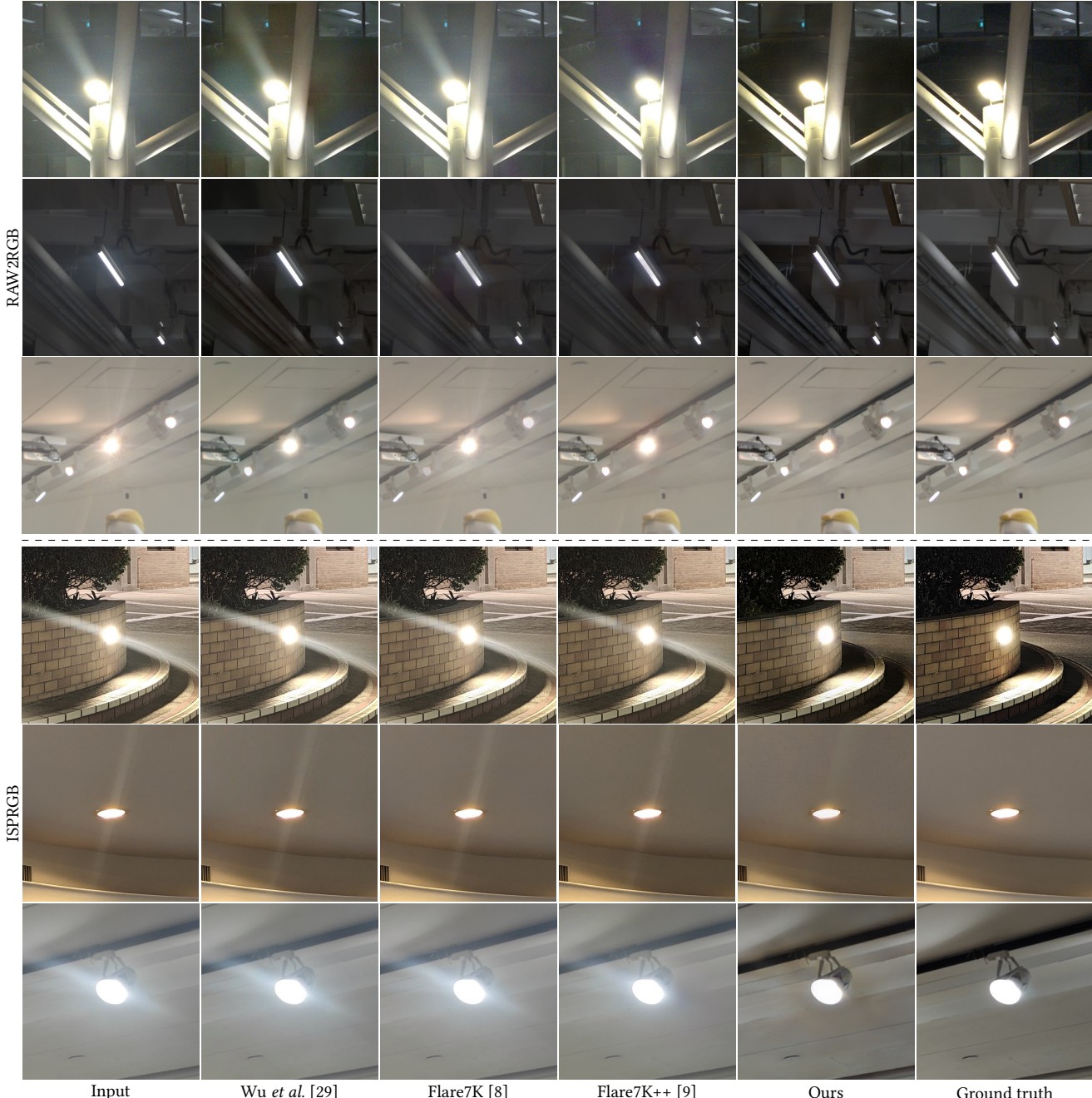

**Figure 6: Comparison of scattering flare removal using different schemes.**

*4.2.2 Qualitative Comparison* We first evaluate the performance of recent flare removal methods on both ISPRGB and RAW2RGB data for reflective and scattering flare images. We observe that these models generally perform better on RAW2RGB images due to their higher quality.

For reflective flares, the U-Net model from Wu *et al.* [29] exhibits difficulties in accurately classifying the flare region for restoration, as shown in Fig. 5. This issue arises from the model's training data,

which primarily consists of scattering flares and lacks sufficient reflective flare examples. Additionally, light-source blending approaches, which segment all light sources and perform flare removal and image blending to merge the results back into the segmented images, may misclassify flares, leading to visual artifacts. For instance, in the second row of Fig. 5, Wu *et al.* [29] misclassify a portion of the cloud as a flare, resulting in inaccurate color restoration. The Uformer model from Flare7K [8], trained with synthetic

**Table 3: Quantitative results on reflective and scattering flare removal for the two types of data. PSNR, SSIM [27], and LPIPS [31] are used for evaluation. ↑ denotes higher is better, and ↓ denotes lower is better.**

| Data | Flare Type | Metric | Input | Wu *et al.* [29] | Flare7K [8] | Bracket Flare [10] | Flare7K++ [9] | Ours |
|------|-----------|--------|-------|------------------|-------------|--------------------|---------------|------|
| RAW2RGB | Reflective | PSNR↑ | 34.034 | 25.810 | 30.356 | 34.172 | – | **35.742** |
| | | SSIM↑ | 0.944 | 0.834 | 0.927 | 0.924 | – | **0.950** |
| | | LPIPS↓ | 0.046 | 0.152 | 0.073 | 0.136 | – | **0.035** |
| | Scattering | PSNR↑ | 20.776 | 22.673 | 21.400 | – | 22.881 | **26.678** |
| | | SSIM↑ | 0.688 | 0.722 | 0.692 | – | 0.701 | **0.749** |
| | | LPIPS↓ | 0.215 | 0.250 | 0.209 | – | 0.190 | **0.145** |
| ISPRGB | Reflective | PSNR↑ | 31.800 | 31.265 | 32.334 | 32.158 | – | **33.611** |
| | | SSIM↑ | 0.956 | 0.953 | 0.955 | 0.938 | – | **0.959** |
| | | LPIPS↓ | 0.050 | 0.056 | 0.051 | 0.090 | – | **0.040** |
| | Scattering | PSNR↑ | 16.558 | 16.774 | 16.982 | – | 17.224 | **20.556** |
| | | SSIM↑ | 0.557 | 0.722 | 0.561 | – | 0.566 | **0.648** |
| | | LPIPS↓ | 0.324 | 0.323 | 0.318 | – | 0.308 | **0.230** |

reflective and scattering flare data, also suffers from this problem, but it is improved in Bracket Flare [10] due to an enhanced training pipeline. However, due to the lack of out-of-focus data in Bracket Flare [10], it can identify the in-focus area to some extent but struggles to resolve the transparent, out-of-focus reflective region, as highlighted in the third row of Fig. 5. Our proposed model, which incorporates both in-the-focus and out-of-focus data for training, demonstrates better performance in addressing this issue.

For scattering flares, recent models demonstrate the ability to restore local scattering flares, as shown in the third and fifth row of Fig. 6. However, these models struggle to handle global flares that affect the overall contrast of images, as seen in the first and last rows of Fig. 6, which are also common in daily life. The visual results from our model shows that, by introducing both local and global degradation data in the training process, we improve the generalization performance of existing approaches.

*4.2.3 Quantitative Comparison* We utilize peak signal-to-noise ratio (PSNR), structural similarity index measure (SSIM) [27], and learned perceptual image patch similarity (LPIPS) [31] for quantitative comparisons. The results presented in Table 3 indicate that the overall image quality obtained with the internal ISP is lower than that of images processed using an external pipeline on raw data. Furthermore, these metrics highlight our proposed approach's enhanced generalization capabilities, particularly in handling global artifacts in scattering flare data and improving clarity in out-of-focus areas for reflective flares.

## 4.3 Investigating Critical Processing Steps

*4.3.1 Experiment Settings* Reflecting on the discrepancy in restoration performance between images directly converted from raw data (RAW2RGB) and those processed through an ISP (ISPRGB), this study further investigates the influence of specific processing steps such as denoising, sharpening, and JPEG compression. We systematically introduce these steps into the RAW2RGB pipeline to evaluate their cumulative effect on image quality. The experiment employs the following processing steps:

- **Denoise:** Denoising in image processing removes noise—caused by sensor flaws, poor lighting, or high ISO settings—to enhance

image clarity and quality, which is often positioned as the initial step. In the experiments, denoising is performed using the medium-sized NAFNet [7] model, which is trained on the SIDD [1] dataset and optimized for a compromise between denoising efficacy and computational efficiency.

- **Sharpen:** The sharpening step is typically placed at the end of an ISP pipeline to enhance the perceptual quality of images by increasing the visibility of edges and details that may have been softened during earlier processing stages. To approximate the varied sharpening approaches used by different smartphone manufacturers, a USM (Unsharp Masking) sharpening operator with a dynamic range of weights is utilized.

- **Compression:** Acknowledging the widespread use of JPEG compression in mobile imaging, the pipeline incorporates DiffJPEG [23] for simulating compression. The quality factor in image compression, particularly in formats like JPEG, balances compression rate and image quality, where a higher factor preserves more details and increases file size, while a lower factor enhances compression efficiency at the expense of data loss. In the experiments, we use quality levels variably set between medium ranges.

To dissect the impact of these processing steps on flare removal efficacy, we explore four distinct pipeline configurations, each altering the sequence of flare removal in relation to denoising, sharpening, and compression:

(1) *Denoise → Flare Removal → Sharpen → Compression:* This setup prioritizes noise reduction before addressing any other image artifacts.

(2) *Denoise → Sharpen → Flare Removal → Compression:* Here, flare removal is executed after sharpening but before the final compression, differing from traditional arrangements.

(3) *Denoise → Sharpen → Compression → Flare Removal:* Mimicking a common practice, this configuration applies flare removal after the complete ISP sequence, which is typical of most existing approaches.

(4) *Sharpen → Compression → Flare Removal:* Designed to replicate scenarios with incomplete noise reduction, possibly due to computational limitations or low-light imaging conditions, this setup omits the initial denoising step.

**Table 4: Impact of processing operations on image restoration performance on reflective flare. PSNR, SSIM [27], and LPIPS [31] are used for evaluation. ↑ denotes higher is better, and ↓ denotes lower is better.**

| Operations | | | Metrics | | | | | |
|---|---|---|---|---|---|---|---|---|
| Denoise | Sharpen | Compression | PSNR (dB) ↑ | Δ(dB) | SSIM↑ | Δ | LPIPS↓ | Δ |
|  |  |  | 35.742 | − | 0.954 | − | 0.035 | − |
| ✓ |  |  | 37.130 | +1.388 | 0.988 | +0.035 | 0.023 | -0.012 |
| ✓ | ✓ |  | 35.902 | -1.228 | 0.986 | -0.002 | 0.027 | +0.004 |
| ✓ | ✓ | ✓ | 34.201 | -1.701 | 0.968 | -0.018 | 0.099 | +0.072 |
|  | ✓ | ✓ | 33.719 | -2.023 | 0.955 | +0.001 | 0.139 | +0.104 |

**Table 5: Impact of processing operations on image restoration performance on scattering flare.**

| Operations | | | Metrics | | | | | |
|---|---|---|---|---|---|---|---|---|
| Denoise | Sharpen | Compression | PSNR (dB) ↑ | Δ(dB) | SSIM ↑ | Δ | LPIPS ↓ | Δ |
|  |  |  | 26.678 | − | 0.749 | − | 0.145 | − |
| ✓ |  |  | 28.111 | +1.433 | 0.943 | +0.194 | 0.070 | -0.075 |
| ✓ | ✓ |  | 26.959 | -1.152 | 0.924 | -0.019 | 0.078 | +0.008 |
| ✓ | ✓ | ✓ | 26.361 | -0.598 | 0.905 | -0.019 | 0.128 | +0.050 |
|  | ✓ | ✓ | 25.701 | -0.977 | 0.842 | +0.093 | 0.193 | +0.048 |

*4.3.2   Results and Analysis*  Analyzing the results compiled in Tables 4 and 5, several critical observations emerge:

**Denoising Benefit:** Echoing previous findings, initiating the pipeline with denoising consistently enhances image quality across various metrics and both types of flare. This confirms the pivotal role of noise reduction in improving overall image clarity and facilitating more effective flare removal. Notably, the fourth configuration underscores the challenges posed by incomplete noise reduction, common in devices with constrained processing power or images taken under poor lighting, leading to significant quality degradation.

**Sharpening and Compression Degradation:** The integration of sharpening and compression steps, regardless of their sequence, invariably diminishes image quality. This outcome highlights the delicate balance between the desire to enhance visual sharpness and the adverse effects of artifact introduction and loss of detail. The impact varies with the type of flare, with reflective flare being particularly vulnerable due to its localized nature, complicating the model's ability to accurately identify and correct affected regions, resulting in up to a 2.93dB degradation in performance. Conversely, scattering flares, which exhibit more global characteristics, face a different set of challenges. The global degradation effects caused by sharpening and compression mirror those inherent in scattering flares, resulting in a performance loss of 1.75dB.

**Optimal Flare Removal Placement:** The analysis suggests that positioning flare removal after denoising but before sharpening and compression (as in configuration 1) tends to produce the best restoration outcomes in terms of PSNR, SSIM, and LPIPS for both flare types. This arrangement benefits from a cleaner input image, enhancing the efficacy of flare removal. Placing flare removal later in the pipeline, especially after sharpening and compression (configurations 2 and 3), appears less effective, likely due to the compounded artifacts and information loss from preceding steps.

The examination of these pipeline configurations illuminates the complex interplay between different ISP processing steps and their collective impact on the quality of image restoration. It confirms that while denoising significantly improves image quality, subsequent processing steps like sharpening and compression can introduce negative effects that compromise both objective and perceptual quality measures. This comprehensive analysis not only clarifies the performance disparity observed between RAW2RGB and ISP-processed images but also offers actionable insights for optimizing ISP pipelines to achieve superior image restoration results.

## 5   Conclusion

This paper tackled the significant challenge of lens flare in mobile computational photography by introducing a novel raw image dataset tailored for the analysis and removal of both scattering and reflective flares. By utilizing RAW data, we can investigate those non-invertible ISP operations, and provide new and critical insights for addressing the flare problem on real data, which is not feasible with previous methods. Through rigorous experimentation, we've illuminated the complex effects of various non-invertible Image Signal Processing (ISP) operations—namely denoising, sharpening, and compression—on the performance of flare removal algorithms. We've provided a nuanced understanding of how different ISP operations impact flare removal, particularly emphasizing the importance of denoising as a foundational step for enhancing image quality. This study's insights into the interplay between ISP steps and flare removal offer valuable guidance for optimizing image processing pipelines, ultimately facilitating better image restoration techniques and improving the visual quality of photographs captured with mobile devices. Our findings reveal the delicate balance between enhancing image quality and preserving essential details necessary for effective flare mitigation, highlighting the critical role of processing sequence in achieving optimal restoration results.

## Acknowledgments

This research has been supported by a donation from SmartMore Corporation.

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
