# OpenReview forum: "Understanding and Tackling Scattering and Reflective Flare for Mobile Camera Systems"
_acmmm.org/ACMMM/2024/Conference — MM2024 Poster_

### Official Review · Reviewer_6KXu · 2024-05-10

**Rating:** 3
**Confidence:** 4

**Summary:**

The paper proposes the first raw image dataset for lens flare removal. The dataset contains lots of real-world flare-corrupted and clean image pairs. Furthermore, the paper investigates the impact of non-invertible ISP operations (denoising, compression, sharpening) on flare removal.

**Strengths:**

1. The paper introduces a novel, high-quality raw image dataset for lens flare removal which contains lots of flare-corrupted and clean image pairs captured in real world.
2. The paper explores the impact of non-invertible ISP operations on flare removal, facilitating the task of flare removal with raw image data.

**Limitations:**

1. From experimental results, the paper archives the best results. However, since other compared methods are trained based on different datasets with domain gaps, is such a comparison fair?
2. For the exploration of the Impact of non-invertible ISP operations, the experimental results is interesting. However, the analysis did not provide an explanation for the causes within. Can the authors provide a more persuasive analysis of the reasons?
3. From Fig. 2 and Fig. 3, different ISP processes have a significant impact on the final JPEG results. For flare removal, as shown in Table 3, your ISP pipeline reaches a better result compared to the internal pipeline. However, does your ISP pipeline have an impact on other aspects of image quality?
4. The way of utilizing raw data for flare removal seems to lack novelty.

**Suitability:**

3

---

### Official Review · Reviewer_fhQE · 2024-05-16

**Rating:** 5
**Confidence:** 3

**Summary:**

This paper introduces a novel approach to addressing scattering and reflective flares in mobile camera systems. It presents a comprehensive raw image dataset with over 2,000 scattering flare and 1,200 reflective flare images captured across diverse scenarios. By focusing on the effects of ISP operations such as denoising, sharpening, and compression on flare removal, the research identifies how these processes can either aid or obstruct flare mitigation. Detailed experiments demonstrate the varying impacts of these ISP steps, offering insights for optimizing ISP configurations.

**Strengths:**

1. Provide a detailed examination of the impact of scattering and reflective flares on mobile camera image quality.

2. Offer critical insights into how different ISP operations, such as denoising, sharpening, and compression, influence flare removal.

3. Introduce a large-scale, high-quality dataset specifically tailored for studying and mitigating flare effects in diverse real-world scenarios.

**Limitations:**

1. The paper does not address the full range of commonly used ISP operations on mobile devices, such as color correction, white balancing, and HDR processing, potentially limiting the comprehensiveness of the proposed flare removal solutions.

2. The paper does not provide a detailed evaluation of the computational efficiency and real-time performance of the proposed flare removal methods, which are important considerations for practical implementation on mobile devices.

3. The paper does not specify whether the introduced dataset will be made publicly available, which could hinder community-driven advancements and independent validation of the proposed methods.

**Suitability:**

2

---

### Official Review · Reviewer_S2aU · 2024-05-22

**Rating:** 3
**Confidence:** 3

**Summary:**

This paper introduced a novel dataset tailored for the analysis and removal of both scattering and reflective flares to solve the problem of lens flare in mobile computational photography. The extensive experiments shows its effectiveness.

**Strengths:**

1. This paper proposed a high-quality raw image dataset for lens flare removal.
2. This paper investigated the impact of non-invertible ISP operations (denoising, compression, sharpening) on flare removal.
3. Enhances the generalizability of existing flare removal approaches by including both local and global flare corruptions data from the dataset.

**Limitations:**

1. I think the authors should add a section to elaborate on the differences between the proposed dataset and existing flare datasets, such as [1][2]
[1] Flare7K: A Phenomenological Nighttime Flare Removal Dataset
[2] How to train neural networks for flare removal
2. In Chapter 4.1, how does the author determine the position of the light source through light source detection? What is the method of light source detection?
3. In the experimental section, why different methods are used for experiments on different datasets? I think this is unfair to other methods.
4. Reference formats need to be consistent, such as conference/journal abbreviations.
5. The author should provide detailed experimental settings, such as GPU type, image size during training, etc.

**Suitability:**

3

---

### Official Review · Reviewer_PEY8 · 2024-05-24

**Rating:** 4
**Confidence:** 4

**Summary:**

This paper introduces a new raw image dataset tailored to eliminate flare for mobile camera systems. Experimental analysis supports the claim and results look significant.

**Strengths:**

1.This paper proposes a high-quality raw image dataset for lens flare removal.

2.The motivation of the article is to incorporate the flare removal step into ISP for optimizing ISP pipelines and effectively solving lens flare, which is attractive and meaningful and may inspire other works.

3.Writing is good and the paper is easy to follow and understand.

**Limitations:**

1.In the process of data capture, different devices may bring different visual effects for the same scene. How to ensure the optimal quality of data and reduce data discrepancies?

2.Lack of detailed experiments on out of distribution and real-world image generalization.

3.What would be the effect if the flare removal step is placed after the denoising step? It is recommended to further supplement experiments to demonstrate the impact of ISP operation on flare removal.

4.Suggest further explaining the differences between the proposed dataset and other datasets.

**Suitability:**

2

---

### Meta-Review · Area_Chair_CC4i · 2024-07-02

**Recommendation:** Accept (Poster)
**Confidence:** 5

**Metareview:**

After rebuttal, all the 4 reviews are positive, so I recommend the acceptance of this paper.